# Impact of *FLT3*-ITD Insertion Length on Outcomes in Acute Myeloid Leukemia: A Propensity Score-Adjusted Cohort Study

**DOI:** 10.3390/biology11060916

**Published:** 2022-06-15

**Authors:** Elizabeth M. Corley, Moaath K. Mustafa Ali, Hanan Alharthy, Kathryn A. F. Kline, Danielle Sewell, Jennie Y. Law, Seung Tae Lee, Sandrine Niyongere, Vu H. Duong, Maria R. Baer, Ashkan Emadi

**Affiliations:** 1University of Maryland School of Medicine, Baltimore, MD 21201, USA; elizabeth.corley@som.umaryland.edu; 2University of Maryland Greenebaum Comprehensive Cancer Center, Baltimore, MD 21201, USA; moaath.mustafaali@umm.edu (M.K.M.A.); kathryn.kline@umm.edu (K.A.F.K.); jennielaw@umm.edu (J.Y.L.); seunglee@umm.edu (S.T.L.); sandrine.niyongere@umm.edu (S.N.); vduong@umm.edu (V.H.D.); mbaer@umm.edu (M.R.B.); 3Department of Medicine, University of Maryland School of Medicine, Baltimore, MD 21201, USA; halharthy@som.umaryland.edu; 4Translational Genomics Laboratory, University of Maryland Greenebaum Comprehensive Cancer Center, Baltimore, MD 21201, USA; dsewell@som.umaryland.edu; 5Department of Pharmacology, University of Maryland School of Medicine, Baltimore, MD 21201, USA

**Keywords:** acute myeloid leukemia, survival_3_, prognosis_4_, FLT3-ITD

## Abstract

**Simple Summary:**

Acute myeloid leukemia (AML) is a complicated disease with well-studied genetic mutations that have been used in both risk stratification as well as targets for treatment. In this study, we examined the varying prognostic significance of different lengths of the special mutation, named internal tandem duplication or ITD, of a commonly mutated gene in AML named FLT3. We found that the longer the size of the ITD mutation, the better the clinical outcomes.

**Abstract:**

The prognostic significance of the length of internal tandem duplication (ITD) insertions in mutant *FLT3* genes in acute myeloid leukemia (AML) is controversial. We conducted a retrospective study to evaluate the correlation between the ITD base-pair (bp) insertion length and clinical outcomes. The mutational status of the *FLT3* gene was evaluated in 402 of 467 consecutive AML patients treated at the University of Maryland Greenebaum Comprehensive Cancer Center between 2013 and 2020; 77 had *FLT3*-ITD mutations. Patients were divided into three cohorts based on bp insertion length (<30 (0–33rd percentile), 30–53 (34th–66th percentile),and >53 (>66th percentile)). The median overall survival (OS) of patients was 16.5 months (confidence interval (CI) 7.3-NA), 18.5 months (CI 7.3-NA), and 21.9 months (CI 19.1-NA) (*p* = 0.03) for the <30, 30–53, and >53 bp insertion length cohorts, respectively. The adjusted median event-free survival (EFS) for the ITD insertion lengths >30, 30–53, and >53 bp was 11.1 months (CI 2.8–16.5), 5.2 months (CI 2.9–12.6), and 9.1 months (CI 5.4-NA) (*p* = 0.5), respectively. Complete remission (CR) rates were 64% (<30 inserted bp), 55% (30–53 inserted bp), and 79% (>53 inserted bp) (*p* = 0.23). For patients treated with gilteritinib and midostaurin, the unadjusted median OS was not statistically significantly different between cohorts.

## 1. Introduction

Mutations in the gene encoding the FMS-like tyrosine kinase 3 (*FLT3*) receptor tyrosine kinase occur in approximately 30% of acute myeloid leukemia (AML) cases, with internal tandem duplication (ITD) representing the most common type (25% of AML cases) [1]. Guidelines from organizations such as the World Health Organization (WHO), National Comprehensive Cancer Network (NCCN), and European LeukemiaNet (ELN) for AML classification and risk stratification rely heavily on cytogenetic aberrations and gene mutations [1,2,3,4]. *FLT3*-ITD mutations are associated with high leukemic burden and poor prognosis in AML [5]. ITDs occur in the juxtamembrane domain of the FLT3 receptor. They cause disruption of the FLT3 signaling, which results in the loss of its autoinhibitory function and leads to FLT3 ligand-independent phosphorylation and activation of the FLT3 receptor [6,7]. *FLT3*-ITD mutations are heterogenous in allelic burden, size, and location [8].

Currently, utilizing polymerase chain reaction (PCR) to amplify the 328 bp region within the *FLT3* gene is achieved by using target-specific fluorescently labeled primers. This amplification method and fragment-size detection on the ABI 3730 genetic analyzer and analysis using GeneMappet software allows detection of the length of the insertions as well as the mutant allele burden. Currently, there are conflicting data about the relationship between the size of the *FLT3*-ITD insertion length and clinical outcome. *FLT3*-ITD insertion size does not currently affect clinical decision-making. Retrospective studies have variably shown worse overall survival (OS) with increasing insertion length [9,10,11,12,13,14], no impact on OS [15,16], poorer survival with medium-size ITD length [17], better OS with increased ITD insertion length [18], and differences by domain of the mutation [19]. Stirewalt et al. suggested [9] that the increasing ITD insertion length leads to a greater loss of the autoinhibitory function of FLT3, thus conferring higher risk with longer insertion; however, Kusec et al. [18] asserted that insertion length might not be related directly to the activation of these signaling pathways. In this study, we aimed to further elucidate the role of the *FLT3*-ITD base-pair (bp) insertion length in AML prognostication in a propensity score-adjusted retrospective study. We also aimed to investigate the correlation of the ITD insertion length with response to the FLT3 inhibitors midostaurin and gilteritinib. 

## 2. Methods

### 2.1. Study Design

We conducted a single-site retrospective cohort study to compare OS, event-free survival (EFS), complete remission (CR), and complete remission with incomplete hematologic recovery (CRi) rates in adults with AML with different *FLT3*-ITD lengths from 2013 (the beginning of in-house PCR testing for *FLT3* mutations at our institution) to 2020. All patients tested for *FLT3*-ITD mutation length using PCR were included. Patients not tested for *FLT3*-ITD mutations were excluded.

Treatment response was evaluated in accordance with the 2017 ELN criteria [11]. Composite CR rate (CCR) included CR+CRi. EFS was defined as the time from treatment initiation to induction failure, relapse, or death from any cause. OS was defined as the time from diagnosis to death from any cause. The study was approved by the University of Maryland Baltimore (UMB) Institutional Review Board (IRB). 

### 2.2. Analysis of FLT3 Mutation

Genomic DNA was extracted and amplified by PCR using primers targeted to the *FLT3* gene that can identify ITD variants in exons 14 and 15. The wild-type *FLT3* allele is detected as a 328 base-pair PCR product. The insertion length is determined by subtracting the base-pair size of the mutant peak from the base-pair size of the wild-type peak. The sensitivity threshold for detecting a *FLT3*-ITD variant is 3.1% allele burden relative to that of the normal DNA sequence, as per University of Maryland Translational Genomics laboratory validation protocols. NCBI reference sequences are NM_004119.2, NP_004110.2, and NG_007066.1. In order to determine the insertion site locations for comparison, we collected available Next-Generation Sequencing (NGS) data and mapped ITD mutations to domains by amino acid: juxtamembrane (JM), hinge region, and tyrosine kinase domain (TKD) I and II. 

### 2.3. Data Source

We reviewed the medical records of patients diagnosed with AML at the University of Maryland Greenebaum Comprehensive Cancer Center (UMGCCC) (2013–2020) using the electronic medical records systems, including Epic and its features such as Care Everywhere and CRISP, a state-designed Health Information Exchange for a Maryland online database [20]. Data were collected and managed using Research Electronic Data Capture (REDCap) electronic data capture tools hosted at the University of Maryland [21,22]. 

### 2.4. Variables and Comparison Groups

Data extracted included age, gender, ethnicity, Eastern Cooperative Oncology Group (ECOG) performance status (PS), baseline comorbidities, AML categories (de novo, myelodysplasia-related, myeloproliferative-related, therapy-related), cytogenetics, available myeloid mutations, therapy received, and clinical outcomes. We compared patients with different ITD insertion length mutations. We compared insertion sites in different domains: JM, hinge region, and TKD I and II. Data were checked three times by independent data collectors. Patients were divided into three cohorts by percentiles: 0–33rd percentile (bp insertion < 30), 34–66th percentile (bp insertion 30–53), and >66th percentile (bp insertion > 53). These cut-offs were chosen to analyze the length as a continuous variable for a dose–response relationship, similar to previous studies [9,10,13,15].

### 2.5. Propensity Score Estimation

As we previously described [23], we conducted propensity score modeling to adjust for outcomes. This study obtained the Average Treatment Effect (ATE) as an estimand. We included the following variables in the propensity score model: age at diagnosis, gender, ethnicity, comorbidities, ECOG-PS, type of AML, cytogenetics at diagnosis, and *ASXL1*, *FLT3, RUNX1,* and *TP53* mutational status. Inverse probability weighting method was used for the propensity score models. We used different methods to estimate weights, including multinomial regression, generalized boosted modeling, covariate balancing, and non-parametric covariate balancing [24]. The choice of weighting method was based on achieving the lowest standardized biases differences, smallest coefficients of variations, and largest estimated sample size [24]. Standardized bias score < 0.25 was used as a cutoff for model choice [25]. We used balance tables and Love plots to assess for covariate balance before and after matching. Robust variance estimator was used to account for within-person homogeneity [26]. Patients with relapsed or refractory AML receiving single-agent gilteritinib were analyzed without propensity score estimation. 

### 2.6. Statistical Analysis

We used descriptive statistics to compare baseline characteristics of patients with various length *FLT3*-ITD mutations as well as *FLT3*-WT. Categorical variables were presented as absolute numbers and percentages. Continuous variables were presented as medians with interquartile ranges (IQR) or means with standard deviations (SD). Baseline characteristics were compared using t-test or analysis of variance (ANOVA) when continuous or Pearson chi-square or Fisher’s exact test when categorical. OS and EFS were compared using log-rank and Gehan–Breslow–Wilcoxon rank tests. Multivariable and univariable Cox proportional hazards models (CPH) were used to assess relative mortality. To test for effect modification of *NPM1* mutation on outcomes, we estimated OS in ITD insertion categories, stratified by *NPM1* mutational status. Moreover, we estimated a multivariable CPH model to predict mortality with *NPM1* mutational status and an interaction term between *NPM1* and ITD insertion length added to the model. We used regression diagnostics to evaluate model assumptions. All statistical tests were two-sided, and *p*-values < 0.05 were considered statistically significant. The R-statistical package “WeightIT” was used for propensity score weighting [25,26]. R-statistical software (version 4.1.1) was used for statistical analyses.

## 3. Results

### 3.1. Cohort Characteristics

A total of 467 AML patients were treated at UMGCCC during the study period (2013–2020). The *FLT3* mutational status was evaluated in 402 patients. *FLT3*-ITD mutations were present in the AML cells of 77 patients (19%). The median age of the total cohort was 65.3 years [IQR 54.2–74.8], and 200 (44%) were female. The median follow-up for the whole cohort was 76.5 months (confidence interval (CI) 23.9–102.9). The median follow-up was 27 months (CI 13.2–76.5) for the <30 bp group, 29.3 months (CI 26-NA) for the 30–53 bp group, and 80.1 months (CI 78-NA) for the >53 bp group. The comparison groups included 25 patients with a <30 bp insertion, 28 patients with a 30–53 bp insertion, and 24 patients with a >53 bp insertion. Table 1 shows the baseline characteristics of patients with *FLT3*-ITD, categorized by insertion length. After weighting, all standardized biases scores were less than <0.25 (Appendix A).

### 3.2. Outcomes of FLT3-ITD-Mutated AML, Categorized by Base-Pair Insertion Length

On the weighted propensity score analysis, the adjusted CCR rates for the 73 treated patients with insertion lengths <30, 30–53, and >53 bp were 64, 55, and 79%, respectively, *p* = 0.23. The covariate balances before and after propensity score weighing for OS are shown in Appendix A. The adjusted median OS between groups increased with the bp insertion length: 16.5 months (CI 7.3-NA) for the <30 bp cohort compared to 18.5 months (CI 7.3-NA) for the 30–53 bp cohort and 21.9 months (CI 19.1-NA) for the >53 bp cohort (*p* = 0.03). The unadjusted median OS for the three cohorts was 13.8 months (CI 7.2-NA), 12.1 months (CI 7.3-NA), and 20.6 months (CI 19.1-NA) (*p* = 0.2), respectively. Compared to the <30 bp insertion cohort, the weighted mortality hazard ratios (HR) for the bp 30–53 and bp > 53 cohorts were 0.79 (CI 0.37–1.65; *p* = 0.52) and 0.56 (CI 0.29–1.09; *p* = 0.09), respectively. On the unweighted multivariable Cox proportional hazard, patients with an insertion length of >53 bp had a relative mortality of 0.44 (CI 0.24–0.81, *p* = 0.009) and those with an insertion length of 30–53 bp had a relative mortality of 0.98 (CI 0.55–1.75; *p* = 0.94) compared to patients with an insertion length <30 bp. Figure 1 demonstrates propensity score-adjusted OS for patients in all three cohorts. The adjusted OS at years 1 and 2 for all three cohorts are shown in Appendix A.

The EFS was assessed in 73 of the 77 patients with *FLT3*-ITD mutations who received treatment. The covariate balances before and after propensity score weighing for the EFS are shown in Appendix A. The adjusted median EFS for the *FLT3*-ITD mutation insertion lengths >30, 30–53, and <53 bp were 11.1 months (CI 2.8–16.5), 5.2 months (2.9–12.6), and 9.1 months (5.4-NA), respectively, (*p* = 0.5). The adjusted EFS at years 1–3 for patients in all three insertion length cohorts showed no statistically significant differences (Appendix A). Compared to patients with insertion lengths of <30 bp, the relative mortality or progression were not statistically significantly different in patients with an insertion length of 30–53 bp (HR 1.22, CI 0.69–2.17, *p* = 0.5) or >53 bp (HR 0.70, CI 0.35–1.39, *p* = 0.31), using the weighted univariable Cox proportional hazards regression. Figure 2 demonstrates the propensity score-adjusted EFS for patients in all three bp insertion lengths cohorts. 

### 3.3. Outcomes of FLT3-ITD-Mutated and Wild-Type (WT) AML, Categorized by Insertion Length

To compare the outcomes of insertion lengths categories with *FLT3*-WT AML, regardless of other cytogenetics or mutational status, another inverse probability weighting model was estimated. The unadjusted median OS for patients with *FLT3*-WT versus *FLT3*-ITD AML (lengths: <30, 30–53, >53 bp) was 12 months (CI 10.2–15.9) compared to 16.5 months (CI 12.5-NA), 21.8 months (CI 9.8-NA), and 23.6 months (CI 20.2-NA) (*p* < 0.001). The covariate balances before and after propensity score weighing for OS are shown in Appendix A. Appendix A demonstrates the propensity score-adjusted OS for patients with *FLT3*-ITD AML versus *FLT3*-WT AML. 

### 3.4. Outcomes of FLT3-ITD-Mutated AML Categorized by Domain Insertion Expansion Categories

Clinical outcomes were analyzed for patients for whom NGS data were available according to ITD insertion site location. Twenty-five patients had insertions in the JM domain, three patients had insertions in the hinge region, and six patients had insertions in the TKD domain. The unadjusted median OS for patients with *FLT3*-ITD insertions in the hinge region was 28.1 months (CI 16.8-NA) compared to 18.5 months (CI 12.1-NA) for patients with *FLT3*-ITD insertions in the JM domain versus 20.5 months for patients with *FLT3*-ITD insertions in the TKD domain (CI 9.5-NA) (*p* = 0.6, with the reference as the hinge domain). Appendix A demonstrates the propensity score-adjusted OS for patients with *FLT3*-ITD AML versus *FLT3*-WT AML. Compared to the hinge domain insertion cohort, the weighted mortality hazard ratios for the JM domain and TKD domain cohorts were 1.99 (CI 0.46–8.67; *p* = 0.36) and 1.26 (CI 0.23–6.95; *p* = 0.79), respectively.

### 3.5. Outcomes after Using Midostaurin plus Induction Chemotherapy in FLT3-ITD AML, Categorized by Base-Pair Insertion Length

Patients who received midostaurin plus cytarabine and daunorubicin induction therapy were analyzed according to the ITD bp insertion length. Five patients had an insertion length of <30 bp, five had an insertion length of 30–53 bp, and two had an insertion length of >53 bp. The unadjusted CCR rates for insertion lengths <30, 30–53, and >53 bp were 80%, 100%, and 100% (*p* = 0.56).

The unadjusted median OS was 19.1 months (CI 19.1-NA) for the <30 bp cohort, 21.8 months (CI 21.8-NA) for the 30–53 bp cohort, and not reached for the >53 bp cohort (*p* = 0.8). Appendix A demonstrates OS curves for patients in all three groups. Using linear regression, the average change in the FLT3 allelic burden after midostaurin treatment with induction was −28.3% for the <30 bp cohort (standard error (SE) 28.3%, *p* = 0.16), −49.9% for the 30–53 bp group (SE 25.8%; *p* = 0.44), and −7.29% for the >53 bp group (SE 45%; *p* = 0.65), respectively.

### 3.6. Outcomes after Using Gilteritinib Monotherapy in Relapsed or Refractory FLT3-ITD AML, Categorized by Base-Pair Insertion Length

Patients with relapsed or refractory AML receiving single-agent gilteritinib were analyzed for outcomes according to bp insertion length. Eight patients had a <30 bp insertion length, nine patients had an insertion length of 30–53 bp, and nine patients had an insertion length of >53 bp. The unadjusted CR+CRi rates were 88%, 67%, and 44%, respectively (*p* = 0.18). The unadjusted median OS in the insertion length groups was 29.9 months (CI 16.3-NA) for the <30 bp cohort, 18.9 months (CI 4.7-NA) for the 30–53 bp cohort, and 11.7 months (CI 5.13-NA) for the >53 bp cohort; however, this finding was not statistically significant (*p* = 0.3). Appendix A demonstrates unadjusted OS curves for patients with relapsed or refractory AML treated with gilteritinib.

Using linear regression, the average change in the *FLT3* allelic burden after the initial use of gilteritinib treatment was −37.4% for the <30 bp cohort (SE 17.5%; *p* = 0.054), −6.64% for the 30–53 bp cohort (SE 24.7; *p* = 0.24), and −4.98% for the >53 bp cohort (SE 24.7%; *p* = 0.21), respectively.

### 3.7. Outcomes for Patients with FLT3-ITD and Mutated Nucleophosmin (NPM1) AML

We tested outcomes for *NPM1*-mutated compared to *NPM1* wild-type AML with different ITD lengths and the results are presented in Appendix A and Appendix A. We excluded patients who did not have their *NPM1* status checked. There were 26 patients with both *NPM1* and *FLT3*-ITD mutations: 9 in the <30 bp cohort, 9 in the 30–53 bp cohort, and 8 in the >53 bp cohort. There were 23 controls (*NPM1* WT with *FLT3*-ITD mutation). The median OS of patients in different insertion length groups are presented in Appendix A, stratified by the *NPM1* mutational status. The *NPM1* mutational status was not associated with a statistically significant change in relative mortality (Appendix A). In the Cox proportional hazards model shown in Appendix A, we added an interaction term between the ITD bp insertion length, modeled as a continuous variable, and the *NPM1* mutational status. The interaction coefficient indicates that *NPM1*-WT was associated with higher mortality when the length of the ITD insertion increased; however, this was not statistically significant. 

In the Cox proportional hazards model shown in Appendix A, we added an interaction term between the ITD bp insertion length, modeled as a categorical variable, and the *NPM1* mutational status. The interaction coefficient indicates that *NPM1*-WT is associated with higher mortality when the length of the ITD was >53 bp and lower mortality when the length was between 30 and 53 bp, both compared to length <30 bp; however, this was not statistically significant. 

## 4. Discussion

Prognostic models for AML primarily rely on cytogenetic aberrations and molecular abnormalities, such as *FLT3*-ITD, *NPM1,* and *CEBPA* mutations in patients with a normal karyotype [27]. Here, we aimed to elucidate the prognostic significance of a commonly reported laboratory finding: *FLT3*-ITD insertion size in AML with this molecular abnormality. When patients were divided into three cohorts based on bp insertion length percentile, we found that the adjusted median OS improved with a longer bp insertion length. This finding remained consistent on analysis when we compared the patient outcomes of all three *FLT3-ITD* length cohorts to *FLT3*-WT. This may be due to a high proportion of poor-risk features, such as complex karyotype and *TP53* mutations in AML with *FLT3*-WT. As shown in Table 1, none of the *FLT3*-ITD groups had high-risk cytogenetics. 

Previous studies have theorized that increasing the insertion size leads to a greater loss of autoinhibitory function of the FLT3 receptor, thus conferring higher risk with longer insertion [9,10,11,12]. These studies have hypothesized that small ITDs may better preserve the autoinhibitory function of the juxtamembrane domain, whereas larger ITDs completely disrupt this function, leading to uninhibited proliferation. Our study, using propensity score adjustment, suggests that size may play a different role than previously thought. It could be hypothesized that the size of the ITD insertion plays a role in autoinhibitory downstream effects and is not directly tied to disruption. One study reported no prognostic significance of size, but rather that mutations that occurred more toward the 5′ end were associated with a better outcome than mutations closer to the 3ʹ end [16]. This also suggests that perhaps location plays a role in autoinhibitory function. Our study directly identified a larger size as more favorable. Upon review of the effect of the insertion location, we observed that the closer the expansion to the membrane, the poorer the survival. However, we only had small numbers of patients with this sequencing information, underpowering the results, which was statistically insignificant. It is possible that location along the domain and size may be more directly linked than previously thought, and in vitro studies examining correlations between size and insertion location may further elucidate the complex interaction between the two. In a brief report, Kayser et al. showed similar findings to ours, using a multivariate Cox proportional hazard model that demonstrated that increasing ITD size was associated with a better prognosis [14]. 

The overall survival for patients with relapsed or refractory disease treated with gilteritinib decreased with an increasing insertion length: 29.9 months (CI 16.3-NA) for the <30 bp cohort, 18.9 months (CI 4.7-NA) for the 30–53 bp cohort, and 11.7 months (CI 5.1-NA) for the >53 bp cohort. This trend may be explained by better mechanistic inhibition of *FLT3*-ITD with the shorter insertion length by gilteritinib, with less efficacy in long insertions. To date, the Phase 3 clinical trial of giltertinib in relapsed or refractory FLT3-mutated AML has not reported any negative prognostic effect of the insertion length on survival outcomes [27]. The efficacy of giltertinib against relapsed or refractory AML with certain base insertion lengths is independent of the prognostic significance of the insertion length in the total *FLT3*-mutated population of this study. Though these results were not statistically significant, the small sample size and thus power may have led to a Type II statistical error. 

Upon testing for an *NPM1* and *FLT3*-ITD length interaction, we found that *NPM1* did not predict outcomes independently and did not modify the effect of bp insertion lengths on outcomes. This was either because of our small number of patients (N = 26), or because though the two have previously been shown to be associated [28,29,30,31,32], the *NPM1* mutation may not be biologically affected by different FLT3 insertion lengths. One other retrospective study has shown a consistent positive prognostic effect of *NPM1* mutations regardless of the *FLT3*-ITD mutation status [15]. 

The major limitation of this study is that it was a retrospective single-site model with a limited number of patients. Because retrospective studies are inherently confounded, we used propensity score weighting, weighted multivariable regression, and unweighted multivariable regression to control for observable confounding. After weighting, the standardized mean difference was less than 0.2 in all variables. 

## 5. Conclusions

Though previous studies have been conflicting, we found that increased *FLT3*-ITD length was associated with better outcomes in a propensity score-adjusted cohort. Future studies, both laboratory and clinical, should be performed to elucidate the link between *FLT3*-ITD size aberration and outcome as a means of enhancing prognostication and clinical decision-making.

## Figures and Tables

**Figure 1 biology-11-00916-f001:**
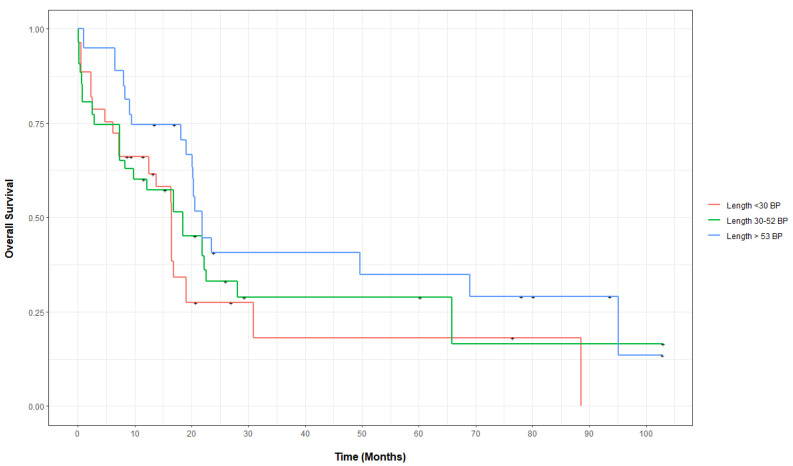
Propensity score-adjusted overall for patients with *FLT3*-ITD-mutated AML, categorized by insertion length.

**Figure 2 biology-11-00916-f002:**
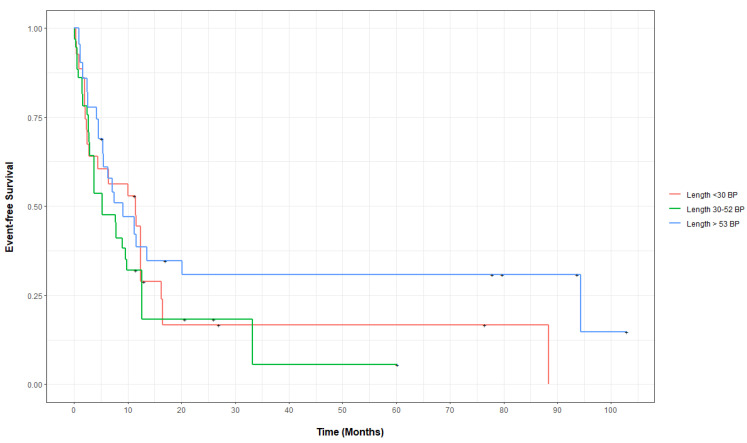
Propensity score-adjusted event-free survival for patients with *FLT3*-ITD-mutated AML, categorized by insertion length.

**Table 1 biology-11-00916-t001:** Baseline characteristics of patients with *FTL3*-ITD mutations, categorized by base-pair insertion length.

	<30 bp	Percentage/SD/IQR	30–53 bp	Percentage/SD/IQR	>53 bp	Percentage/SD/IQR	*p*-Value
Number of patients	25	-	28	-	24	-	-
Age (Average ± SD)	63.9	18.40	63.8	15.2	58.6	15.2	0.43
Age (Median, IQR)	66.9	58–79.3	65.3	56.9–72.7	60.8	45.9–71.1	0.60
Female	5	20	12	43	17	71	0.002
Blood Counts at Diagnosis							
White Blood Cells (K/microL) (Median, IQR)	18.5	8.8–79.9	75.9	21–105.6	70.7	13.2–130.3	0.03
Hemoglobin (g/dL) (Median, IQR)	8.4	7.3–9.6	8.6	7.1–9.3	8.2	7.3–9.5	0.75
Platelets (K/microL) (Median, IQR)	59.0	24–113	70.5	34.5–119	48.0	29.75–75	0.66
Blast percentage (%) (Average ± SD)	55.5	23.4	67.9	26.0	63	23.3	0.56
Body Mass Index	27.0	4.4	25.7	5.3	30	12.8	0.55
Ethnicity							
Causian	18	72.0	20	71.4	17	70.8	0.99
Other	7	28.0	8	28.6	7	29.2	
Comorbidities							
Cardivascular disease	6	24	4	14	6	25	0.57
Diabetes mellitus	7	28	4	14	5	20	0.47
Hypertension	11	44	11	39	9	38	0.89
CKD stage III-V/ESRD	1	4	2	7	1	4	0.84
Active Cancer	0	0	1	3.6	0	0	0.41
AML type							0.37
AML, de novo	17	68	22	78.6	19	79.2	
AML with MDS/CMML changes	8	32	4	14.3	4	16.7	
Therapy-Related AML	0	0	2	7.1	1	4.2	
ELN 2017 Cytogenetic Category							0.680
Favorable Risk	1	4	2	7.1	0	0	
Intermediate Risk	22	88	25	89.3	22	91.7	
Unfavorable Risk	0	0	0	0	0	0	
Not performed/Poor banding, Inadequate	2	8	1	3.6	2	8.3	
*FLT3*-ITD status							0.37
*FLT3*-ITD-mutated allelic burden 1–49%	18	72	15	53.6	15	62.5	
*FLT3*-ITD-mutated allelic burden 50–100%	7	28	13	46.4	8	33.3	
*FLT3* wild type	0	0	0	0	1	4.2	
*FLT3*-TKD mutated	10	40	4	14.3	7	29.2	0.11
*TP53* mutational status							0.01
*TP53* mutated	1	4	0	0	0	0	
*TP53* wild type	19	76	18	64.3	11	45.8	
*TP53* untested	5	20	10	35.7	13	54.2	
*RUNX1* mutational status							0.07
*RUNX1* mutated	4	16	2	7.1	0	0	
*RUNX1* wild type	16	64	16	57.1	11	45.8	
*RUNX1* untested	5	20	10	35.7	13	54.2	
*ASXL1* mutational status							0.02
*ASXL1* mutated	3	12	0	0	0	0	
*ASXL1* wild type	17	68	18	64.3	11	45.8	
*ASXL1* untested	5	20	10	35.7	13	54.2	
*NPM1* mutational status							0.08
*NPM1* mutated	9	36	9	32.1	8	33.3	
*NPM1* wild type	11	44	9	32.1	3	12.5	
*NPM1* untested	5	20	10	35.7	13	54.2	
*CEBPA* mutational status							0.1
*CEBPA* mutated	1	4	4	11	1	4	
*CEBPA* wild type	19	76	15	53	10	42	
*CEBPA* untested	5	20	10	36	13	54	
ECOG-PS status							0.15
I or II	23	92	23	82.1	24	100	
III or IV	2	8	4	14.3	0	0	
ECOG status unknown	0	0	1	3.6	0	0	
Types of first-line treatment							0.28
Anthracycline-based	10	40	12	42.9	16	66.7	
Non-anthracycline-based	13	52	14	50	8	33.3	
None	2	8	2	7.1	0	0	
Midostaurin with induction	5	20	5	17.8	2	8.3	0.48

## Data Availability

The data that support the findings of this study are available from MMA upon reasonable request.

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
