# Peer review of "Impact of FLT3-ITD Insertion Length on Outcomes in Acute Myeloid Leukemia: A Propensity Score-Adjusted Cohort Study"

_biology, 2022, doi:10.3390/biology11060916_

Round 1

Reviewer 1 Report

The manuscript describes the prognostic significance of ITD insertions in the mutant FLT3 gene in AML. Authors have conducted retrospective study to establish a correlation between ITD base pair insertion length and clinical outcomes. However, the major limitation of this study is the retrospective single-site model with a limited number of patients, which the authors have addressed. The study offers some advancement in the field of FLT3-ITD AML prognosis. The discussion could be elaborated. Overall, the work is good, but the authors should consider evaluating additional datasets because of the small sample size and the results being non-statistically significant.

Include the changes directly marked in yellow after the incorrect version and included after >. They should be copied and contained at the respective site. 

Major issues:

  1. The authors have not found a statistically significant difference between the OS and EFS of patients with different FLT3-ITD insertion lengths. The authors should consider using the publically available dataset to strengthen their findings.
  2. Authors could carry out an in-vitro study by using a cell line with FLT3-ITD wild-type and mutant form, treating with gilteritinib inhibitor, and checking for the FLT3 -ITD length.
  3. Authors should mention any known effect of type 2 inhibitors in the length of FLT3 -ITD length in the discussion session.
  4. Authors in the discussion should detail why they think adjusted median OS improved with longer bp insertion length associated with TP53 mutation.  
  5. Did the authors check the outcomes of FLT3-ITD with mutated CEBPA AML?

Minor issues:

In introduction: This amplification method along with fragment size detection on the ABI 3730 genetic analyzer and analysis using GeneMappet software allows detection of the length of the insertion as well as the mutant allele burden.

>This amplification method and fragment size detection on the ABI 3730 genetic analyzer and analysis using GeneMappet software allows detection of the insertion length and the mutant allele burden.

Author Response

June 1st, 2022

Re: Revised manuscript Impact of FLT3-ITD Insertion Length on Outcomes in Acute Myeloid Leukemia: A Propensity Score-Adjusted Cohort Study” (Manuscript ID: Biology- 1677146)

Dear Biology Editorial Office Staff,

We would like to thank you and the reviewers for reviewing and considering this manuscript for publication in Biology. We appreciate that you and the reviewers provided constructive criticism and comments. Please find the following revisions in response to the reviewers’ critiques, addressing them point by point.

Per your instruction, the newly added or deleted words are highlighted using the track changes. Here are the details of the changes we made to the original manuscript:

Reviewer 1:

General Comment

  • The manuscript describes the prognostic significance of ITD insertions in the mutant FLT3 gene in AML. Authors have conducted retrospective study to establish a correlation between ITD base pair insertion length and clinical outcomes. However, the major limitation of this study is the retrospective single-site model with a limited number of patients, which the authors have addressed. The study offers some advancement in the field of FLT3-ITD AML prognosis. The discussion could be elaborated. Overall, the work is good, but the authors should consider evaluating additional datasets because of the small sample size and the results being non-statistically significant.
    • Response: Thank you for reviewing and summarizing our manuscript.

Major Issues:

Comment #1:

  • The authors have not found a statistically significant difference between the OS and EFS of patients with different FLT3-ITD insertion lengths. The authors should consider using the publically available dataset to strengthen their findings..
    • Response: This is an excellent idea as an additional piece of information for the project. We have performed a thorough search and unfortunately we were not able to find any public database that reports information on the base insertion length of ITD mutations in FLT3 mutant AML. Future plans is to collaborate with other institutions and combine data.

Comment #2:

  • Authors could carry out an in-vitro study by using a cell line with FLT3-ITD wild-type and mutant form, treating with gilteritinib inhibitor, and checking for the FLT3-ITD length.
    • Response: We appreciate this suggestion. The in vitro activity of gilteritinib against AML cell lines with and without FLT3-ITD mutations have already been reported [See: Mori M, Kaneko N, Ueno Y, et al. Gilteritinib, a FLT3/AXL inhibitor, shows antileukemic activity in mouse models of FLT3 mutated acute myeloid leukemia. Invest New Drugs. 2017;35(5):556-565] and reviewed comprehensively [See: Levis M, Perl AE. Gilteritinib: potent targeting of FLT3 mutations in AML. Blood Adv. 2020 Mar 24;4(6):1178-1191.] We also have performed the suggested experiments and presented it at the American Society of Hematology (ASH) 60th Annual Meeting [See: Sintim HO, Aman MJ, Holtsberg F, Emadi A, Lapidus, RG. KRX-101, a novel FLT3 inhibitor, potently active against resistant FLT3-ITD/FLT3-TKD mutant AML in vitro and in vivo. American Society of Hematology (ASH) 60th Annual Meeting, San Diego, CA; December 2018; Abstract 4049.]. We think it is neither necessary nor germane to our manscript to repeat the results of those already published data.

Comment #3:

  • Authors should mention any known effect of type 2 inhibitors in the length of FLT3-ITD length in the discussion session.
    • Response: Thank you for the recommendation. We have added the following sentence to the third paragraph in the discussion to address this point: “To date, the Phase 3 clinical  trial of giltertinib in relapsed or refractory FLT3 mutated AML has not reported any negative prognostic effect of insertion length on survival out-comes (30).”

Comment #4:

  • Authors in the discussion should detail why they think adjusted median OS improved with longer bp insertion length associated with TP53 mutation.
    • Response: Thank you for the comment. We have not analyzed any outcome associations with  TP53 mutation, rather, included it in our propensity score analysis.

Comment #5:

  • Did the authors check the outcomes of FLT3-ITD with mutated CEBPA AML?
    • Response: Per Reviewer’s suggestion, we have added CEPBA mutated AML to our propensity score analysis and the results have not changed. The results is reported in the new Table 1.

Minor Issues:

Comment #6:

  • In introduction: This amplification method along with fragment size detection on the ABI 3730 genetic analyzer and analysis using GeneMappet software allows detection of the length of the insertion as well as the mutant allele burden.

> This amplification method and fragment size detection on the ABI 3730 genetic analyzer and analysis using GeneMappet software allows detection of the insertion length and the mutant allele burden.

    • Response: Thank you for this comment. We have substituted the original sentence with this new sentence in the Introduction to read: “This amplification method and fragment size detection on the ABI 3730 genetic analyzer and analysis using GeneMappet software allows detection of the length of the insertions as well as the mutant allele burden.”

Reviewer 2 Report

The main finding is that the length of the ITD- fragment has prognostic role and the larger the ITD fragment, especially > 53 mb. This matter is still controversial and the findings of the study are in contrast to many of the previous studies which showed that the length of the fragment has no prognostic role. The  manuscript is well-written and  based on a retrospective study of 72 patients with FLT3-ITD mutated AML from the same center during several years. Furthermore, the authors did the propensity score matching in order to match three groups of patients. It is It would be very interesting to know if ITD insertion site in the beta1-sheet TKD1 610-615 had a prognostic importance.  It would be nice to show the map of mutation within the FLT3 gene as it was in reference 14.  ITD-length is showed previously to be strongly correlated with functional regions of the gene. It would be good that the authors explain why they used 3 values of ITD-fragment why not 2 values or one cutoff-value, because <30 and 30-53 are within the average length of the fragment. Did the authors tried to do some other comparisons not counting percentiles.

The treatment could have prognostic significance. Maybe I missed to see if the patients were uniformly treated, if everyone got Midostaurin or not. If not, the results could be unprecise. 

Both conclusions are counter-intuitive: the larger the fragment, better the prognosis and better effect of gilteritinib. Even though there are similar findings it does not make sense, and better explanation of the findings is necessary in order to complete the discussion. 

Author Response

June 1st, 2022

Re: Revised manuscript Impact of FLT3-ITD Insertion Length on Outcomes in Acute Myeloid Leukemia: A Propensity Score-Adjusted Cohort Study” (Manuscript ID: Biology- 1677146)

Dear Biology Editorial Office Staff,

We would like to thank you and the reviewers for reviewing and considering this manuscript for publication in Biology. We appreciate that you and the reviewers provided constructive criticism and comments. Please find the following revisions in response to the reviewers’ critiques, addressing them point by point.

Per your instruction, the newly added or deleted words are highlighted using the track changes. Here are the details of the changes we made to the original manuscript:

Reviewer 2:

Comment #1:

  • The main finding is that the length of the ITD-fragment has prognostic role and the larger the ITD fragment, especially > 53 mb. This matter is still controversial and the findings of the study are in contrast to many of the previous studies which showed that the length of the fragment has no prognostic role. The  manuscript is well-written and  based on a retrospective study of 72 patients with FLT3-ITD mutated AML from the same center during several years. Furthermore, the authors did the propensity score matching in order to match three groups of patients. It would be very interesting to know if ITD insertion site in the beta1-sheet TKD1 610-615 had a prognostic importance. It would be nice to show the map of mutation within the FLT3 gene as it was in reference 14. ITD-length is showed previously to be strongly correlated with functional regions of the gene.
    • Response: Thank you for your valuable comments. We believe that addressing these comments have improved the scientific quality of our manuscript. Per Reviewer’s suggestion, we investigated ITD insertion site effect on prognosis. We looked at patients for whom in addition to PCR for FLT3 identification, Next Generation Sequencing (NGS) were also performed and identified 37 subjects and compared outcomes based on different domains in FLT3 gene. In order to maintain statistical power, we grouped the patients into Juxtamembrane Domain (JM), Hinge Region, and TKD (including both TKD1 and TKD2). In summary, the closer the expansion to the membrane, the shorter the survival, albeit not statistically significant probably due to a small sample size. We also performed a Cox regression with correction to age and the conclusion was the same. We only had two patients with mutations in the beta1-sheet (610-615) so we could not compare that group directly.
    • We added the following sections to our methods:

2.2 – last sentence: “In order to determine the insertion site locations for comparison, we collected available Next Generation Sequencing (NGS) data and mapped ITD mutations to domains: Juxtamembrane (JM), Hinge Region, and Tyrosine Kinase Domain (TKD) I & II.”

2.4 – middle of the section: “We compared insertion sites in different domains: JM, Hinge Region, and TKD I & II.” 

    • We added the following section to our results:

3.4. Outcomes of FLT3-ITD mutated AML categorized by domain insertion expansion categories

Clinical outcomes were analyzed for patients for whom NGS data were available according to ITD insertion site location. Twenty five patients had insertions in the JM domain, three patients had insertions in the Hinge region, and six patients with insertions in the TKD domain. The unadjusted median OS for patients with FLT3-ITD insertions in the Hinge region was 28.1 months (CI 16.8-NA) compared to 18.5 months (CI 12.1-NA) for patients with FLT3-ITD insertions in the JM domain, versus 20.5 months for patients with FLT3-ITD insertions in TKD domain (CI 9.5-NA) (P=0.6, with reference as Hinge domain). eFigure 5 demonstrates propensity score-adjusted OS for patients with FLT3-ITD AML versus FLT3-WT AML. Compared to the Hinge domain insertion cohort, the weighted-mortality hazard ratio for the JM domain and TKD domain cohorts was 1.99 (CI 0.46-8.67; P=0.36) and 1.26 (CI 0.23-6.95; P=0.79), respectively.”

    • We also added to the discussion, referencing this finding:

“Upon review of the effect of insertion location, we observed that the closer the expansion to the membrane, the poorer the survival. However, we only had small numbers of patients with this sequencing information, underpowering the results which was statistically insignificant.”

Comment #2:

  • It would be good that the authors explain why they used 3 values of ITD-fragment why not 2 values or one cutoff-value, because <30 and 30-53 are within the average length of the fragment. Did the authors tried to do some other comparisons not counting percentiles.
    • Response: Thank you for your careful review. To treat insertion length as a continuous variable, we chose greater than one single cut off. Ideally, quartiles would have improved this process; however, with only 72 patients, each group would be limited in size. Therefore, three groups was ideal for analyzing for continuous variable while maintaining power in each group.

Comment #3:

  • The treatment could have prognostic significance. Maybe I missed to see if the patients were uniformly treated, if everyone got Midostaurin or not. If not, the results could be unprecise.
    • Response: Thank you for the great point. We had 12 patients treated with Midostaurin, details of which we have added to Table 1.  We have now also included a section in the results discussing Midostaurin effect modification on outcomes, please see Results section 3.5.
    • 3.5 Outcomes after using midostaurin plus induction chemotherapy in FLT3-ITD AML, categorized by base-pair insertion length

Patients who received midostaurin plus cytarabine and daunorubicin induction therapy were analyzed according to ITD bp insertion length. Five patients had insertion length <30 bp, five insertion length 30-53 bp, and two insertion length >53 bp. The unadjusted CCR rates for insertion lengths <30 bp, 30-53 bp, and >53 bp were 80%, 100%, and 100% (P=0.56).

The unadjusted median OS was 19.1 months (CI 19.1-NA) for the <30 bp cohort, 21.8 months (CI 21.8-NA) for the 30-53 bp cohort, and not reached for the >53 bp cohort (P=0.8). eFigure 5 demonstrates OS curves for patients in all three groups. Using linear regression, the average change in FLT3 allelic burden after midostaurin treatment with induction was -28.3% for <30 bp cohort (standard error (SE) 28.3%, P=0.16), -49.9% for 30-53 bp group  (SE 25.8%; P=0.44), and -7.29% for the >53 bp group  (SE 45%; P=0.65), respectively.”

Comment #4:

  • Both conclusions are counter-intuitive: the larger the fragment, better the prognosis and better effect of gilteritinib. Even though there are similar findings it does not make sense, and better explanation of the findings is necessary in order to complete the discussion.
    • Response: Thank you for your comment; it is an interesting point. We found that patients with longer insertion sites had better overall outcomes, and though statistically insignificant, also had worse response to Gilteritinib. We believe that both can in fact be true, as the majority of patients are not treated with Giltertinib and further, treatment was accounted for in our propensity score analysis. Nonetheless, we have addressed this point in the third paragraph of the discussion now:

“The efficacy of giltertinib against relapsed or refractory AML with certain base insertion lengths is independent of the prognostic significance of insertion length in the total FLT3-mutated population of this study.”

Round 2

Reviewer 1 Report

The authors have carried out all the corrections and I recommend the manuscript to be considered for the publication. 

Reviewer 2 Report

No further comments. The authors have answered correctly to my queries.